

# Abelian combinatorial gauge symmetry

Hongji Yu[1], Dmitry Green[1,2], Andrei E. Ruckenstein[1] and Claudio Chamon[1]

**1** Physics Department, Boston University, Boston, MA, 02215, USA
**2** AppliedTQC.com, ResearchPULSE LLC, New York, NY 10065, USA

## Abstract

Combinatorial gauge symmetry is a principle that allows us to construct lattice gauge theories with two key and distinguishing properties: a) only one- and two-body interactions are needed; and b) the symmetry is *exact* rather than emergent in an effective or perturbative limit. The ground state exhibits topological order for a range of parameters. This paper is a generalization of the construction to *any* finite Abelian group. In addition to the general mathematical construction, we present a physical implementation in superconducting wire arrays, which offers a route to the experimental realization of lattice gauge theories with static Hamiltonians.

doi:10.21468/SciPostPhysCore.7.1.014

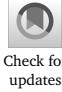
# 1 Introduction

Physically realizing topological ordered [1] gapped quantum spin liquids in the laboratory remains a problem at the frontier of our knowledge in both quantum materials and quantum information sciences. Control over such states would constitute a major step forward in building stable quantum memories, material simulation, and ultimately scalable quantum computation. Over the past decades, major theoretical advances have been made by formulating models within the framework of lattice gauge theories [2], for example the well-known $\mathbb{Z}_2$ toric code [3]. Recently, $\mathbb{Z}_2$ quantum spin liquids have been demonstrated by preparing a state like the quantum ground state of the static system and its elementary excitations as non-equilibrium states in quantum simulators [4,5]. However, as these states are prepared dynamically, they do not exist in the absence of a an external drive or modulation, and thus they do not possess the topological stability of the ground states of the static Hamiltonians.

One challenge with physical realization of quantum spin liquids in the static setting is that most well-understood theoretical models require multi-body (e.g., 4- or 6-spin) interactions. (A notable exception is Kitaev's honeycomb model [6].) While it is possible to design multi-spin interactions as effective couplings arising from physical, two-body interactions, these constructions tend to be either perturbative (they are exact in a classical limit) and break down in the presence of quantum dynamics [7–16], or require capabilities beyond those achievable in real materials or in programmable quantum devices [17–21].

A set of proposals utilizing the "combinatorial gauge symmetry" (CGS) construction plots a potential alternative path to overcoming these obstacles [22, 23]. CGS is a framework for constructing Hamiltonians out of realistic two-body interactions where the gauge symmetry is *exact and non-perturbative* for the full range of parameters. As such there is hope that a suitable range of parameters can be found where the topological phase is sufficiently gapped to be observable and stable. Earlier work has centered on special cases of small Abelian groups ($\mathbb{Z}_2$ and $\mathbb{Z}_3$). By generalizing the framework to all Abelian groups we further enlarge the scope of possible implementations and parameters where one can look for such states. The mathematical construction developed in this paper to handle the generalization is interesting in its own right.

From a mathematical point of view, central to the construction is a classification of two-body interactions that can be written as matrices $W_{ij}$ that couple two quantum variables indexed by $i$ and $j$ (e.g., spins or Josephson phases). The allowed sets of interactions $W$ must possess several properties in order to satisfy the requirements of a lattice gauge theory exactly. This paper is a systematic exposition of the general requirements.

As a high level preview, the gauge symmetry arises from automorphisms of the form $W = L^\top W R$ where the matrices $L$ and $R$ must be monomials in order to preserve the commutation relations of the underlying quantum variables. Further, because of the geometry of the lattice configuration, $R$ will be required to be diagonal, but $L$ will not. Effectively, the problem of classifying the allowed $W$ matrices will reduce to a study of how group actions relate to permutations of invariant sets.

The outline of the paper is as follows. First we introduce in Sec. 2 a simple example of a model with a combinatorial gauge symmetry corresponding to a $\mathbb{Z}_2$ gauge theory on the triangular lattice to illustrate key elements of our construction. We summarize the main results of this paper in Sec. 3. In Sec. 4 we develop in detail the general construction for any Abelian gauge group, as well as modifications to the models in specific cases, and then provide illustrations for these in Sec.5. Finally, we discuss experimental implementations in Sec. 6.

## 2 A motivating example

In this section we consider a $\mathbb{Z}_2$ gauge theory on a triangular lattice with only two-body interactions. Before that, it should be made clear what kind of gauge symmetry such a theory possesses. In an ordinary $\mathbb{Z}_2$ gauge theory, e.g., the classical toric code model, the degrees of freedom are located on the links of the lattice and the Hamiltonian is

$$H = -\lambda_B \sum_p B_p \,, \tag{1}$$

where the plaquette operator $B_p$ is the product of $\sigma^z$ operators acting on the links bounding the plaquette $p$. In the full toric code model, this Hamiltonian has gauge symmetries generated by star operators, that is, products of $\sigma^x$ operators along a path through the dual lattice that forms a domain wall around a subset of vertices on the lattice. At the level of individual degrees of freedom, such symmetries act by the transformation $\sigma^z \mapsto \sigma^x \sigma^z \sigma^x = -\sigma^z$. Thus, whenever the support of a plaquette operator intersects with that of a gauge transformation, the degrees of freedom involved will be affected by the gauge transformation, but the plaquette operator will be invariant overall. (See Fig. 1(a).) Since in the Abelian case such a symmetry operation is always realized by a phase factor, we may represent its effect on a plaquette term by a diagonal matrix multiplying the vector of operators contained in this term. For example, if we collect the $z$-components of the three spins around a star onto a vector $\boldsymbol{\sigma}^{z\top} \equiv (\sigma_1^z, \sigma_2^z, \sigma_3^z)^\top$, then the action of the symmetry $\mathcal{R}$[1] that flips the first two spins is

$$\begin{pmatrix} \sigma_1^z \\ \sigma_2^z \\ \sigma_3^z \end{pmatrix} \overset{\mathcal{R}}{\mapsto} \begin{pmatrix} -\sigma_1^z \\ -\sigma_2^z \\ \sigma_3^z \end{pmatrix} = R \begin{pmatrix} \sigma_1^z \\ \sigma_2^z \\ \sigma_3^z \end{pmatrix}\,, \tag{2}$$

where $R$ is the diagonal matrix

$$R = \begin{pmatrix} - & & \\ & - & \\ & & + \end{pmatrix}\,. \tag{3}$$

Under the action of $\mathcal{R}$ the product $\sigma_1^z \sigma_2^z \sigma_3^z$ is invariant, $(-\sigma_1^z)(-\sigma_2^z)\sigma_3^z = \sigma_1^z \sigma_2^z \sigma_3^z$, because the phase factors introduced by the two spin flips multiply to one.

Having understood the action of the gauge symmetry on individual degrees of freedom and on gauge invariant operators, we now construct a two-body Hamiltonian that possesses the same gauge symmetry. To this end, in addition to the gauge spinoperators $\sigma^z$ as before, we introduce matter spin operators $\mu^z$ associated with each plaquette term in the pure gauge theory. More precisely, in this example we introduce four matter spins $\mu_a^z$, $a = 1, 2, 3, 4$, which we also collect into a vector $\boldsymbol{\mu}^{z\top} \equiv (\mu_1^z, \mu_2^z, \mu_3^z, \mu_4^z)^\top$. We would like the Hamiltonian to (i) have only one- and two-body interactions (here of the $ZZ$ type) and (ii) possess a $\mathbb{Z}_2$ gauge symmetry when acting on the gauge spins. Condition (i) is consistent with an interaction term of the form $-J \sum_{a=1}^4 \sum_{i=1}^3 W_{ai} \mu_a^z \sigma_i^z$. Using the vector notation, this interaction can be written compactly as $(\boldsymbol{\mu}^z)^\top W \boldsymbol{\sigma}^z$. As for condition (ii), when the gauge symmetry acts on the $\sigma_i$'s via $\boldsymbol{\sigma}^z \mapsto R\boldsymbol{\sigma}^z$, the interaction is transformed into $(\boldsymbol{\mu}^z)^\top W R \boldsymbol{\sigma}^z$, which is equivalent to multiplying the coupling matrix $W$ on the right by $R$. If there exists a matrix $L$ that we can multiply on the left of $W$ so that $L^\top W R = W$, then $L$ is the action of the gauge symmetry on $\boldsymbol{\mu}^z$, so that the transformation $(\boldsymbol{\sigma}^z, \boldsymbol{\mu}^z) \mapsto (R\boldsymbol{\sigma}^z, L\boldsymbol{\mu}^z)$ leaves the system invariant. The $L$ and $R$ matrices arise from gauge transformations, which ultimately come from conjugating spins operators by some

---

[1]In this paper we will use different styles of letters to distinguish the abstract gauge symmetry and the matrices that implement these symmetries by left multiplication on a vector of gauge or matter operators. Calligraphic letters will denote the former, Roman letters the latter.

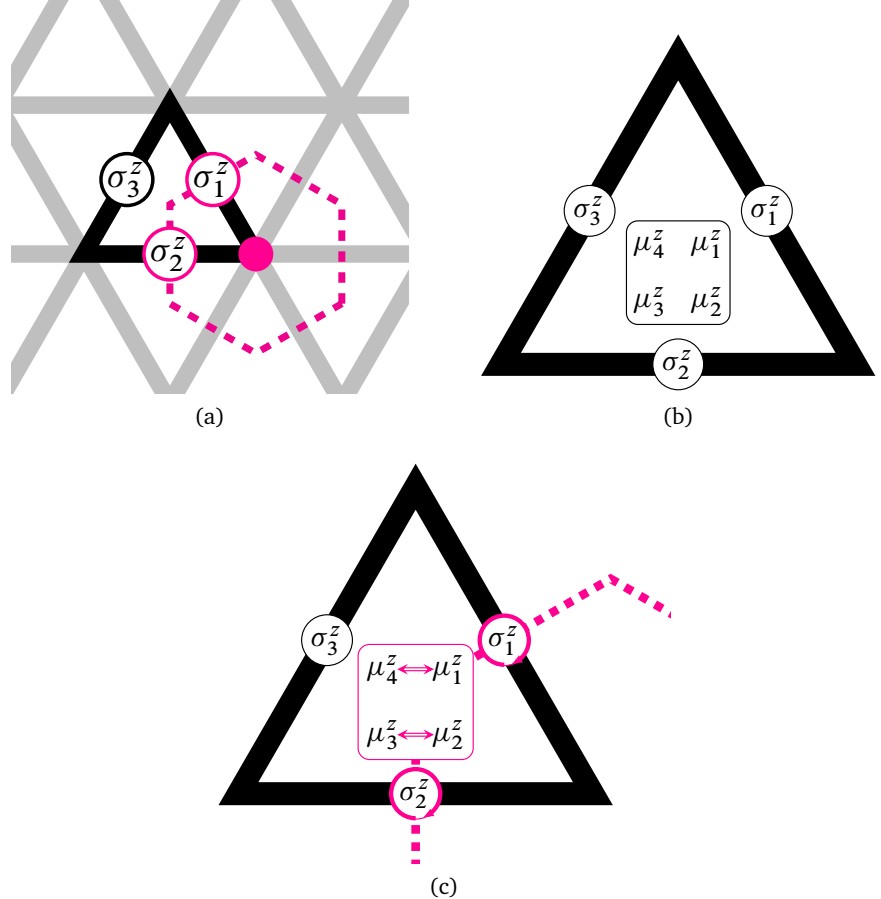

Figure 1: Illustration of the $\mathbb{Z}_2$ gauge symmetry on a triangular lattice. (a) A gauge transformation flips certain spins, but plaquette terms remain invariant overall, since there is always an even number of spin flips. (b) A two-body operator satisfying the combinatorial gauge symmetry corresponding to the gauge symmetry depicted in (a) requires four matter degrees of freedom. (c) Under the action of the gauge symmetry in (a), the combinatorial gauge symmetric operator in (b) stays invariant by flipping the gauge spins just as in the pure gauge theory, while also permuting the matter spins.

symmetry operator, so the transformation represented by $L$ should preserve the commutation relations between the spin operators. Therefore, we restrict our considerations only to $L$'s that permute the spins or flip $\mu^z$'s. In other words, the matrix $L$ should be a monomial matrix with entries $\pm 1$.

In the triangular lattice example, we can construct an interaction that satisfies all of the above conditions with four matter spins. The coefficient matrix is

$$W = \begin{pmatrix} + & + & + \\ + & - & - \\ - & + & - \\ - & - & + \end{pmatrix}.$$ (4)

From the lattice geometry, we see that gauge transformations acting on the gauge spins around

a triangular plaquette are generated by the following two diagonal matrices,

$$
R^{12} = \begin{pmatrix} - & & \\ & - & \\ & & + \end{pmatrix}, \qquad\qquad R^{23} = \begin{pmatrix} + & & \\ & - & \\ & & - \end{pmatrix}. \tag{5}
$$

The superscripts denote the sites where the spins are flipped. The corresponding $L$ matrices that leave $W$ invariant under $W \mapsto L^\top W R$ are

$$
L^{12} = \begin{pmatrix} 0 & 0 & 0 & 1 \\ 0 & 0 & 1 & 0 \\ 0 & 1 & 0 & 0 \\ 1 & 0 & 0 & 0 \end{pmatrix}, \qquad\qquad L^{23} = \begin{pmatrix} 0 & 1 & 0 & 0 \\ 1 & 0 & 0 & 0 \\ 0 & 0 & 0 & 1 \\ 0 & 0 & 1 & 0 \end{pmatrix}, \tag{6}
$$

labeled by the same superscripts as the corresponding $R$ matrices. (See Fig. 1(c) for an illustration of the effect of $R^{12}$ and $L^{12}$ on the operators $\sigma_i^z$ and $\mu_a^z$.) Next we take a closer look at the properties of these matrices.

By design $W$ in (4) has the symmetry $L^\top W R = W$. The rows of $W$ are the triples of $\pm 1$ that multiply to 1. These are precisely the triples that can be obtained from $(+, +, +)$ by flipping the signs of two entries at a time. More formally, the rows of $W$ form the orbit of $(+, +, +)$ under the action of the group of gauge symmetries. Orbits under a group action are examples of invariant subsets, which means that if we flip the sign of any two columns of $W$ via $W \mapsto W R$, i.e. act on all elements of the invariant subset by a group element, the effect is the same as permuting the rows of $W$, i.e. the elements of the invariant subset. Therefore, for each $R$, there will always be a permutation matrix $L$ that "undoes" its effect by permuting the rows back into their original positions, so that $W R = (L^\top)^{-1} W$ is satisfied. For example, with the pair of transformations $R^{12}$ and $L^{12}$, we have

$$
W R^{12} = \begin{pmatrix} + & + & + \\ + & - & - \\ - & + & - \\ - & - & + \end{pmatrix} \begin{pmatrix} - & & \\ & - & \\ & & + \end{pmatrix} = \begin{pmatrix} - & - & + \\ - & + & - \\ + & - & - \\ + & + & + \end{pmatrix}, \tag{7}
$$

and

$$
L^{12} W = \begin{pmatrix} 0 & 0 & 0 & 1 \\ 0 & 0 & 1 & 0 \\ 0 & 1 & 0 & 0 \\ 1 & 0 & 0 & 0 \end{pmatrix} \begin{pmatrix} + & + & + \\ + & - & - \\ - & + & - \\ - & - & + \end{pmatrix} = \begin{pmatrix} - & - & + \\ - & + & - \\ + & - & - \\ + & + & + \end{pmatrix}. \tag{8}
$$

Flipping the signs of all entries in the first two columns of $W$ has the same effect as exchanging the first row of $W$ with the last and the second with the third.

This interaction term needs to be supplemented with additional terms, so the Hamiltonian of the entire system takes the form

$$
H_{\mathrm{CGS}}^{\mathbb{Z}_2} = -J \sum_p (\boldsymbol{\mu}_p^z)^\top W \boldsymbol{\sigma}_p^z - h \sum_p \sum_{a \in p} \mu_a^z - \Gamma \sum_p \sum_{a \in p} \mu_a^x - \Gamma' \sum_{i \in \mathcal{L}} \sigma_i^x , \tag{9}
$$

where the subscripts $p$ on the operators $\boldsymbol{\mu}_p^{x,z}$ and $\boldsymbol{\sigma}_p^{x,z}$ denote the plaquette that the operators are located in, the indices $a \in p$ label the matter spins in plaquette $p$, and the indices $i$ label the gauge spins on the set of links $\mathcal{L}$ of the lattice. If a large transverse field $-\Gamma \sum_p \sum_{a \in p} \mu_a^x$ is applied on the matter spins, it will effectively integrate out the matter degrees of freedom, so that the interaction term $-J \sum_p \left(\boldsymbol{\mu}_p^z\right)^\top W \boldsymbol{\sigma}_p^z$ will perturbatively generate a pure gauge theory. The lowest order term in perturbation theory will be of the form $\sum_p \prod_{i \in p} \sigma_i^z$. In addition, a

star term can also be generated perturbatively through the application of the transverse field on the gauge spins $-\Gamma' \sum_p \prod_{i \in p} \sigma_i^x$, thus fully reproducing the pure gauge theory we started with. We stress that the gauge symmetry of the system described by Eq. (9) is preserved throughout, not merely emergent in a perturbative limit in which we justify integrating out the matter fields, meaning that a topological phase may exist within a large area of the parameter space of couplings $J, h, \Gamma$ and $\Gamma'$.

For this system to be equivalent to a $\mathbb{Z}_2$ gauge theory whose ground state is a spin-liquid state, it is not sufficient to only have the gauge symmetry. We would like ground state in absence of the $\sigma^x$ term to be an eigenstate of the plaquette operators $B_p = \prod_{i \in p} \sigma_i^z$ with eigenvalue 1, because this enforces a zero-flux condition that corresponds to the confined phase in the pure gauge theory.[2] However, in our motivating example on the triangular lattice, the $B_p = 1$ state is always degenerate with the $B_p = -1$ state, because a global spin flip of both matter and gauge spins commutes with the Hamiltonian but anticommutes with $B_p$. In order to split this degeneracy, we apply a field $-h \sum_p \sum_{a \in p} \mu_a^z$ to the matter spins, with the sign of $h$ chosen so that the $B_p = 1$ sector is preferred. Without this term, the ground state manifold is spanned by the flux-0 state $\{\boldsymbol{\sigma}, \boldsymbol{\mu}\} = \{(+, +, +), (+, -, -, -)\}$ and its gauge symmetry partners, along with the flux-1 state $\{(-, -, -), (-, +, +, +)\}$ and its symmetry partners. Since these two sets of states have distinct matter spin magnetization (indeed, the CGS preserves the magnetization of matter spins because it acts as a permutation on the matter spins), we are able to distinguish the flux states by applying a uniform field on the matter spins.

In the above example we outlined how to construct a combinatorial gauge theory whose ground state has topological order. In the following sections we will generalize these ideas.

## 3 Summary of results

The construction presented in the previous section can be repeated in full generality for all finite Abelian gauge groups by considering cyclic gauge groups. This is because any finite Abelian group $G$ can be decomposed into a direct product of cyclic groups $\prod_i \mathbb{Z}_{k_i}$. Focusing on the on the cyclic group $\mathbb{Z}_k$, we work with operators $\sigma^Z$ and $\sigma^X$ acting on a $\mathbb{Z}_k$ clock variable, which satisfy the relation

$$\sigma^X \sigma^Z = e^{2\pi i/k} \sigma^Z \sigma^X. \tag{10}$$

Schematically, the one- and two-body Hamiltonian for such a combinatorial gauge theory takes the following form:

$$H_{\mathrm{CGS}} = -J \sum_p (\boldsymbol{\mu}_p^Z)^\top W \boldsymbol{\sigma}_p^Z - h \sum_p \sum_{a \in p} \mu_a^Z - \Gamma \sum_p \sum_{a \in p} \mu_a^X - \Gamma' \sum_{i \in \mathcal{L}} \sigma_i^X + \mathrm{h.c.}, \tag{11}$$

where we have extended the vector notation of the example of the previous section to write the $ZZ$ term in the first line of Eq. (11), by collecting all $q$ gauge spins and $m$ matter spins on each plaquette into vectors $\boldsymbol{\sigma}^{Z\top} \equiv (\sigma_1^Z, \sigma_2^Z, \dots, \sigma_q^Z)^\top$ and $\boldsymbol{\mu}^{Z\top} \equiv (\mu_1^Z, \mu_2^Z, \dots, \mu_m^Z)^\top$.

The first term in (11) is the gauge-invariant interaction term. A gauge transformation is represented by a pair of matrices $L_p$ and $R_p$ acting on the vectors of operators associated with each plaquette, i.e. $\boldsymbol{\sigma}_p^{Z,X} \mapsto L_p \boldsymbol{\sigma}_p^{Z,X}$ and $\boldsymbol{\mu}_p^{Z,X} \mapsto R_p \boldsymbol{\mu}_p^{Z,X}$. The diagonal $R_p$ matrices are determined by the behavior of the pure gauge theory under gauge transformations, and the $L_p$ matrices are monomial and constructed according to the requirement $L_p^\top W R_p = W$. This last

---

[2]In Abelian gauge theories, the charges and fluxes are dual to each other, and can be labeled by eigenvalues of $B$ operators, or equivalently elements of the gauge group. In this paper we will be refer to the eigenvalues of $B$ operators as "flux", and use the additive representation for the fluxes in the language of modular arithmetic. See also Appendix A for a discussion of the choosing the support of flux vs. charge variables.

requirement ensures the gauge invariance of this interaction term. It is possible to construct the coupling matrix $W$ for any gauge group. Moreover, by exploiting internal symmetries of matter degrees of freedom, the number of matter variables, or equivalently, the size of the matrix $W$ can be reduced. A CGS Hamiltonian on a lattice with coordination number $q$ will have at least $q$ parameters that we will be able to tune to adjust the spectrum, so that the correct flux sector becomes the ground state.

The transverse field $-\Gamma \sum_p \sum_{a \in p} \mu_a^X$ induces dynamics in the matter degrees of freedom. When this term is strong ($\Gamma \gg J$), the matter degrees of freedom are integrated out, and the effective Hamiltonian to the lowest order in perturbation theory is exactly the plaquette term in the pure gauge theory, since that will be the lowest order term in perturbation theory that respects the gauge symmetry. Next if we apply a weak transverse field $-\Gamma' \sum_{i \in \mathcal{L}} \sigma_i^X$, the generators of gauge transformations, that is, the star terms in the pure gauge theory, will be produced. Note that as long as the transverse field on the matter spins is uniform in each plaquette, the gauge symmetry is preserved. Thus the Hamiltonian (11) will always possess an exact gauge symmetry. This means that even though the effective Hamiltonian only becomes exactly equal to the pure gauge theory in the $\Gamma \to \infty$ limit, we should expect the existence of a topological phase that can be connected adiabatically to the confined phase of the pure gauge theory even at finite values of $\Gamma$. Indeed, in the case of $\mathbb{Z}_2$ gauge theory on a square lattice, it has been shown that such a quantum spin liquid phase exists in this limit. [24] This provides evidence that under the $\Gamma' \ll J \ll \Gamma$ energy hierarchy, the topological phase is robust. Since the gap is finite, as long as symmetry breaking perturbations are small compared to this gap, the splitting between the topologically degenerate ground states, or indeed within any flux sector, is exponentially small in the system size, so the topological phase should be preserved. The presence of the term $-h \sum_p \sum_{a \in p} \mu_a^Z$ in Eq. (11) is required to break accidental degeneracies when certain gauge-violating transformations leave the interaction term invariant. This is another measure to ensure the energetic separation of different flux sectors.

## 4   Combinatorial gauge symmetry

Suppose we want to construct a gauge-invariant two-body interaction whose behavior under gauge transformations mirrors that of a pure gauge theory

$$H_{\text{pure}} = -\lambda_A \sum_s A_s - \lambda_B \sum_p B_p \,, \tag{12}$$

where the first group of terms represents charges, i.e., generators of gauge transformations defined on a star labeled by $s$, and the second group enforces the constraint that the ground state is flux-free on each plaquette labeled by $p$. To describe this behavior algebraically, we need to consider three algebraic constructs.

First, the gauge fields can be represented by variables $\sigma$ that take value in the gauge group $G$ and this group naturally acts on these variables.[3] When an element of the gauge group $g \in G$ acts on a gauge variable $\sigma$, we write it schematically as $g \cdot \sigma$.

On the entire lattice, the total Hilbert space is a tensor product of all the local gauge variables $\bigotimes_i \sigma_i$, and is acted on by the direct product of the gauge groups $\prod_i G_i$, where the factor $G_i$ acts only on the variable $\sigma_i$. Not all elements of this direct product are valid gauge transformations, but only the ones that are generated by the charge operators $A_s$. We call this subgroup $\mathcal{G}$ the group of gauge transformations. An operator is gauge invariant if it is invariant under the action of this group. Nevertheless, in order to test if a local operator $\mathcal{O}$ is gauge invariant,

---

[3]More rigorously, the local Hilbert space is isomorphic to the group algebra $\mathbb{C}[G]$, spanned by the group elements as a vector space. It thus carries an action by the gauge group via left multiplication.

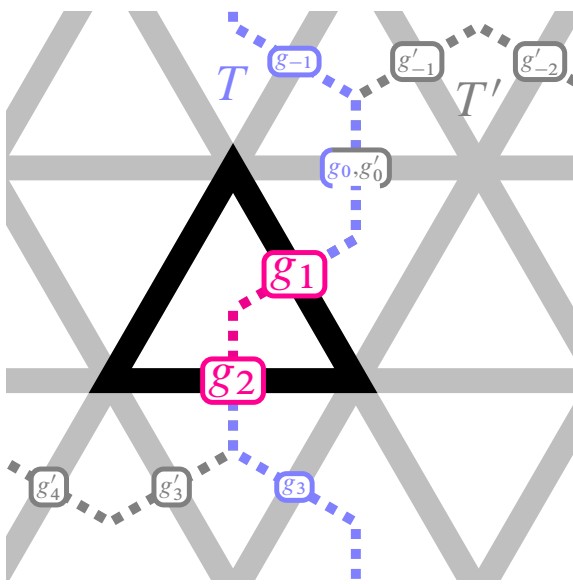

Figure 2: Two gauge transformations $T$ and $T'$ that represent the same element of the local group of gauge transformation (magenta) for a plaquette operator (highlighted in black).

we need not test it against all possible gauge transformations, but only the ones that act non-trivially on the support $S$ of $\mathcal{O}$. This means that we are in fact interested in the quotient group $\mathcal{G}$ under the relation $\sim$ that identifies two gauge transformations $T = \prod_i g_i$ and $T' = \prod_i g_i'$ if $g_i = g_i'$ for all $i \in S$, that is, their actions coincide on the support of the operator $\mathcal{O}$. We define this group as the *local group of gauge transformations* $\mathcal{G}[\mathcal{O}]$ for the operator $\mathcal{O}$. (See Fig. 2 for an illustration of the relation between the group of gauge transformations and the local group of gauge transformations in the $\mathbb{Z}_2$ gauge theory of Section 2.) For compactness, when $\mathcal{O}$ is the flux operator $B_p$, we shall write $\mathcal{G}_p$ instead of $\mathcal{G}[B_p]$.

Now we assume that in the pure gauge theory, the flux operator $B_p$ is a product of $q$ operators $\sigma_1, \ldots, \sigma_q$. However, since our aim is to construct a two-body interaction term involving a set of auxiliary, or "matter", degrees of freedom $\mu_1, \ldots, \mu_m$, the resulting operator should not be a product of all relevant operators. The most natural form is a quadratic direct coupling between the gauge and matter variables, $\sum_{i,a} \mu_a W_{ai} \sigma_i$. Thus we write the gauge and matter degrees of freedom as vectors of operators $\boldsymbol{\sigma}$ and $\boldsymbol{\mu}$, so that the action of a gauge transformation $T \in \mathcal{G}_p$ on $\boldsymbol{\sigma}$ can be represented by a diagonal matrix $R$,

$$\boldsymbol{\sigma} = \begin{pmatrix} \sigma_1 \\ \vdots \\ \sigma_q \end{pmatrix} \overset{T}{\mapsto} \begin{pmatrix} g_1 \cdot \sigma_1 \\ \vdots \\ g_q \cdot \sigma_q \end{pmatrix} = R \begin{pmatrix} \sigma_1 \\ \vdots \\ \sigma_n \end{pmatrix} = R\boldsymbol{\sigma}, \tag{13}$$

where

$$R = \begin{pmatrix} g_1 & & \\ & \ddots & \\ & & g_q \end{pmatrix}, \tag{14}$$

and its action on the matter variables by a matrix $L$, i.e. $\boldsymbol{\mu} \mapsto L\boldsymbol{\mu}$. Now we define that the term $\sum_{i,a} \mu_a W_{ai} \sigma_i = \boldsymbol{\mu}^\top W \boldsymbol{\sigma}$ has a *combinatorial gauge symmetry* that reproduces the gauge symmetry of the flux operator $B_p$ of the pure gauge theory $H_{\text{pure}} = -\lambda_B \sum_p B_p$, if for every gauge transformation $T$ in the local group of gauge transformations $\mathcal{G}_p$ of the flux operator $B_p = \prod_{i \in p} \sigma_i$ that acts on $\boldsymbol{\sigma}_p$ via the matrix $R$, we can find a matrix $L$ acting on a vector of

matter variables $\boldsymbol{\mu}$, such that $(L\boldsymbol{\mu})^\top W(R\boldsymbol{\sigma}) = \boldsymbol{\mu}^\top W\boldsymbol{\sigma}$, or $L^\top WR = W$. The symmetry $L$ of the matter variables should be a physical symmetry of the corresponding physical degrees of freedom.

The overall invariance of $\boldsymbol{\mu}^\top W\boldsymbol{\sigma}$ is equivalent to the invariance of $W$ under the multiplication by $R$ on the right and a corresponding $L$ on the left. Since an $R$ matrix multiplies $W$ on the right, it can be thought of as the same transformation applied to each row of $W$. Thus if the rows of $W$ form an invariant set under all $R$-transformations then multiplying $W$ on the right by a matrix $R$ has the same effect as permuting the rows, so we can "undo" this transformation by multiplying $W$ on the left by a permutation matrix $L^\top$. This gives us the invariance

$$L^\top WR = W, \tag{15}$$

and consequently the invariance of $\boldsymbol{\mu}^\top W\boldsymbol{\sigma}$ under the transformations $\boldsymbol{\mu} \mapsto L\boldsymbol{\mu}$ and $\boldsymbol{\sigma} \mapsto R\boldsymbol{\sigma}$. The number of matter degrees of freedom $\mu$ is then determined by the number of elements in this invariant set. Since invariant sets are unions of orbits, such a coupling matrix $W$ can be constructed by choosing one or more rows of initial entries, applying all $R$ matrix representations of transformations in $\mathcal{G}_p$ to obtain the orbits, and combining the results to form $W$.

We can lay out the construction more concretely using $\mathbb{Z}_k$ as the gauge group. $\sigma$ could be represented by a clock variable $Z = \text{diag}(1, \zeta_k, \zeta_k^2, \ldots, \zeta_k^{k-1})$ where $\zeta_k = e^{2\pi i/k}$. The action of the gauge group is generated by the $X$ operator, associated to a $k \times k$ permutation matrix

$$X = \begin{pmatrix} & 1 & & & \\ & & 1 & & \\ & & & \ddots & \\ & & & & 1 \\ 1 & & & & \end{pmatrix}, \tag{16}$$

that is, take any generator $x$ of the group $G$ and represent it by the $X$ operator, then for any group element $g$, which can be written as $x^c$ for some integer $c$, its action $\sigma \mapsto g \cdot \sigma$ is represented by $Z \mapsto (X^\dagger)^c Z X^c$. In particular, since $XZX^\dagger = \zeta_k Z$, such an action on $Z$ reduces to multiplication by the phase $\zeta_k^c = e^{2\pi ic/k}$. Thus, the action of $\mathcal{G}_p$ that we have represented schematically as a matrix multiplication in fact is a matrix multiplication by a diagonal $R$ matrix whose entries are $k$-th roots of unity. For the flux term $B_p = \prod_{i \in p} X_i$, gauge invariance requires that the entries of $R$ multiply to 1. If $B_p$ is supported on $q$ sites, the local group of gauge transformations $\mathcal{G}_p$ will be a quotient of $\mathbb{Z}_k^{q-1}$. In many cases, $\mathcal{G}_p$ will be precisely $\mathbb{Z}_k^{q-1}$, and when $k$ is prime, $\mathcal{G}_p$ will always be a direct sum of fewer than $q$ copies of $\mathbb{Z}_k$.[4] This includes the case of $\mathbb{Z}_k$ toric codes, but also models with much more complicated geometry, for example the Haah's code, which will be discussed in the following section.

Supposing $\mathcal{G}_p$ is of the form $\mathbb{Z}_k^{q-1}$, we can construct the coupling matrix $W$ by choosing the entries in the first row and generating the other rows under the action of all $R \in \mathcal{G}_p$. Suppose we take the first row of $W$ to be $(1, 1, \ldots, 1)$, then the entries of $W$ will also be $k$-th roots of unity, and the rows of $W$ consist precisely of all $q$-tuples of $k$-th roots of unity that multiply to 1. There are $N = k^{q-1}$ such combinations, so the CGS model will require $k^{q-1}$ matter degrees of

---

[4]Each charge operator will generate a subgroup of the local group of gauge transformation isomorphic to the gauge group. The relation among them is of the form $R_1^{p_1} R_2^{p_2} \ldots R_m^{p_m} = 1$, where $R_i$ is the local gauge transformation generated by the $i$-th charge operator. When $k$ is prime, this relation can be used to express $R_m$ in terms of the other generators, but if not, we may only be able to eliminate a power of $R_m$, resulting in a local group of gauge transformation whose direct summands may include non-trivial quotients of $\mathbb{Z}_k$.

freedom. Explicitly,

$$W^{k,q} = \begin{pmatrix} \zeta_k^{a_{1,1}} & \zeta_k^{a_{1,2}} & \cdots & \zeta_k^{a_{1,q}} \\ \zeta_k^{a_{2,1}} & \zeta_k^{a_{2,2}} & \cdots & \zeta_k^{a_{2,q}} \\ \vdots & \vdots & \ddots & \vdots \\ \zeta_k^{a_{N,1}} & \zeta_k^{a_{N,2}} & \cdots & \zeta_k^{a_{N,q}} \end{pmatrix}, \tag{17}$$

where $a_{n,j}$ are integers such that $\sum_{j=1}^{q} a_{n,j} \equiv 0 \pmod{k}$, or equivalently $\prod_{j=1}^{q} \zeta_k^{a_{n,j}} = 1$, for all $n$. This matrix will satisfy the CGS invariance condition (15). In this case a general gauge transformation acts on the gauge degrees of freedom via a multiplication on the right of $W^{k,q}$ by the following diagonal matrix:

$$R = \begin{pmatrix} \zeta_k^{c_1} & & & \\ & \zeta_k^{c_2} & & \\ & & \ddots & \\ & & & \zeta_k^{c_q} \end{pmatrix}, \tag{18}$$

where $c_j$ are integers satisfying the condition $\prod_{j=1}^{q} \zeta_k^{c_j} = 1$ or $\sum_{j=1}^{q} c_j \equiv 0 \pmod{k}$, so that the product of each row of $W^{k,q}$ is preserved. By construction, this right multiplication on $W$ by $R$ has the effect of permuting its rows, so that an $N \times N$ permutation matrix $L$ exists to rearrange the rows back in their initial order, thereby satisfying the condition $L^\top W R = W$.

For a specific example, consider the $k = 5, q = 3$ case, that is, take $\mathbb{Z}_5$ as gauge group and a triangular lattice. Starting with $(1, 1, 1)$ as the initial row of $W$, the remaining rows can be obtained by the action of the local group of gauge transformations $\mathbb{Z}_5^2$, generated by $R^{12} = \text{diag}(\zeta_5, \zeta_5^{-1}, 1)$ and $R^{23} = \text{diag}(1, \zeta_5, \zeta_5^{-1})$, acting on $W$ by right multiplication, producing a 25-element orbit. (See Appendix B for the full form of this matrix.) In the following discussion we will denote such a combinatorially symmetric coupling matrix by $W^{k,q}[r_1, \ldots, r_n]$, where $r_1, \ldots, r_n$ are representatives of distinct orbits forming the set of its rows, so the $\mathbb{Z}_5$ example above is $W^{5,3}[(1, 1, 1)]$. (When context makes it clear, we may also omit some or all of the arguments of $W$.)

**Reducing the number of matter spins**

In general, we do not impose anything other than permutation symmetry for the matter variables, so in principle the $\mu$'s in the Hamiltonian can be any local operator. However, if the matter variables are realized as spin-$\frac{1}{2}$ or $\mathbb{Z}_n$ degrees of freedom, they could possess an internal symmetry, which generalizes $L$'s to monomial matrices. In some cases, this allows us to reduce the number of matter variables. To be precise, this reduction is possible whenever $\mathcal{G}_p$ contains a subgroup consisting of elements of the form $R = gI$, which we may call "uniform" gauge transformations, forming a subgroup $\mathcal{G}_p^U$ of $\mathcal{G}_p$. Such uniform gauge transformations multiplies all entries by the same group element $g$ from the right and permutes rows of the coupling matrix differing by an overall phase factor from the left. We can factor out all such uniform transformations and let them act as an internal symmetry of the matter variables, or equivalently quotienting $\mathcal{G}_p$ by $\mathcal{G}_p^U$. This reduces the size of $\mathcal{G}_p$ and therefore the number of matter variables by a factor of $\left| \mathcal{G}_p^U \right|$. Starting from a fully constructed CGS model with uniform gauge symmetries, we can partition rows of $W$ into sub-orbits consisting of rows related by a uniform gauge transformation and choose only one representative from each to constructed a reduced $W_{\text{red}}$. For this coupling matrix, when the action of $R$ maps a row of $W_{\text{red}}$ into rows that are not present in the original, it will always be compensated for by a non-identity entry in the $L$ matrix. We will see examples of this below.

## Accidental symmetries

A subtle issue can arise related to uniform gauge transformations. When the matter variables possess internal symmetries, and when such symmetries coincide with elements of the gauge group, they may combine into spurious symmetries of the Hamiltonian that are not members of the local group of gauge transformations. In the language of the coupling matrix $W$ and $L, R$ matrices, this situation is described by a pair of transformations $(L, R) = (g^{-1}I, gI)$, when $gI$ itself is not a valid gauge transformation. For example, this could occur when the matter variables are spin-$\frac{1}{2}$ variables, the gauge group contains an element that is represented by a phase shift by $-1$, and the flux operator $B_p$ contains an odd number of factors. The motivating example discussed in Section 2 falls into this case. Such a spurious symmetry will collapse flux sectors that should remain distinct, so we need to break it with an additional field $-h \sum_p \sum_a \mu_{p,a}^z$ on the matter spins to remove the internal symmetry. (See Appendix C for a demonstration that applying such a field is sufficient to lift the spurious symmetry.)

Note that uniform spurious symmetries arise in a situation complementary to the situation discussed above where uniform gauge transformations allow for a reduction of the coupling matrix, so breaking the internal symmetry in the former case will not come at a cost of increasing the number of matter variables.

## Product gauge groups

The framework for CGS is applicable to general Abelian gauge groups, but it is sufficient to consider the more restricted case of cyclic groups, because any finite Abelian group can be decomposed into a direct product of cyclic groups. A CGS Hamiltonian whose gauge group is a direct product can be built up from CGS Hamiltonians with the factor groups as the gauge group. When the gauge group $G$ is a direct product $G_1 \times G_2$, the group of gauge transformations and the local groups of gauge transformations are also direct products. This means that if $H_1 = -J \sum_p (\mu_p^1)^\top W^1 \sigma_p^1$ has combinatorial gauge symmetry with respect to a $G_1$ gauge theory defined on a lattice and so does $H_2 = -J \sum_p (\mu_p^2)^\top W^2 \sigma_p^2$ with respect to a $G_2$ gauge theory on the same lattice, we can take the tensor product of the two systems, and write

$$H = H_1 \otimes \mathrm{id}_2 + \mathrm{id}_1 \otimes H_2 = -J \sum_p \left[ (\mu_p^1 \otimes \mathrm{id}_2)^\top W^1 (\sigma_p^1 \otimes \mathrm{id}_2) + (\mathrm{id}_1 \otimes \mu_p^2)^\top W^2 (\mathrm{id}_1 \otimes \sigma_p^2) \right],$$
(19)

where $\mathrm{id}_{1,2}$ are identity operators acting on the two tensor sectors. This Hamiltonian will then be a $G_1 \times G_2$ combinatorial gauge theory.

Note that even in cases where the gauge group is already cyclic, the tensor product construct could reduce the number of matter spins. For example, a $\mathbb{Z}_6$ gauge theory on the triangular lattice can be constructed with 36 matter spins, and the number can be reduced to 12 if the matter degrees of freedom have a $\mathbb{Z}_3$ internal symmetry. If the gauge group $\mathbb{Z}_6$ is treated as $\mathbb{Z}_2 \times \mathbb{Z}_3$ so that the gauge theory is constructed through the tensor product construction, only 13 matter spins are needed, with 4 matter spins for the $\mathbb{Z}_2$ sector (see Section 2) and 9 for the $\mathbb{Z}_3$ sector (see Section 5.2). The coupling matrix for the $\mathbb{Z}_3$ sector can be further reduced (also see Section 5.2), so that in total only 7 matter degrees of freedom is required, of which 3 need to be $\mathbb{Z}_3$ variables.

## Ground state flux and tuning the spectrum of the CGS coupling term

We note that there is no fundamental constraint on the initial row(s) of the coupling matrix $W$, so each of these entries can be regarded as a tunable parameter, which gives us a lot of freedom to adjust the spectrum of the CGS coupling term. This is fortuitous, because while the combinatorial gauge symmetry construction guarantees that any model we generate has

an exact gauge symmetry, this does not mean that the ground state of our two-body model also corresponds to the ground state of the gauge theory we are constructing, as we saw in the case of a $\mathbb{Z}_2$ gauge theory on a triangular lattice. To find the energy of a particular flux sector, we can fix a set of value of the $\sigma^Z$ operators that satisfy the flux constraint, then (supposing that the matter variables are spin-$\frac{1}{2}$'s) choose the signs of the $\mu^z$'s, so that the energy is minimized,

$$E_0 = -\sum_a \left| 2\,\mathrm{Re}\left( \sum_i W_{ai}\, \sigma_i^Z \right) \right|. \tag{20}$$

Due to the combinatorial gauge symmetry, this expression only depends on the flux $\prod_{i\in p} \sigma_i^Z$ and the initial coupling values that generate $W$. Given any specific combinatorial gauge symmetry, these initial couplings can be tuned to find a coupling matrix that has the zero flux sector as the lowest energy one. This argument also shows that when the matter degrees of freedom have internal symmetries, transforming them under such symmetries does not change the spectrum of the CGS Hamiltonian. In other words, if the matter degrees of freedom are spin-$\frac{1}{2}$'s, for example, flipping the sign of entire rows of the coupling matrix $W$ will not affect the spectrum.

Take the coupling matrix (17) as an example, if a more general set of coupling constants $(b_1, b_2, \ldots, b_q)$ are taken as the starting point of the construction, the resulting coupling matrix will become $W^{k,q}$ with column $j$ rescaled by $b_j$, and the ground state energy will now become

$$E_0 \mapsto E_0^b = -\sum_a \left| 2\,\mathrm{Re}\left( \sum_i W_{ai}^{k,q}\, b_i\, \sigma_i^Z \right) \right|. \tag{21}$$

In the $W^{5,3}[(1,1,1)]$ example mentioned earlier, the ground state contains both flux-0 and flux-1 and sectors. If instead $r = (-1,1,1)$ or $r = (0.5,1,1)$ is the first row, which corresponds to rescaling the first column of the coupling matrix (40) shown in the appendix by $-1$ or $0.5$, respectively, the ground state will be a flux-0 state.

## 5  Examples of combinatorial gauge symmetric Hamiltonians

In examples to follow, we will examine a few CGS Hamiltonians. Through these examples we shall further illustrate both the flexibility of these models and the necessary conditions for realizing a topological state.

### 5.1  Uniform gauge transformations and reduction of number of matter spins

Consider a $\mathbb{Z}_2$ gauge theory on a square lattice, e.g. the toric code. Using the initial row $(-1,1,1,1)$, we will get the $8 \times 4$ interaction matrix

$$W^{2,4} = \begin{pmatrix} - & + & + & + \\ + & - & - & - \\ + & - & + & + \\ - & + & - & - \\ + & + & - & + \\ - & - & + & - \\ + & + & + & - \\ - & - & - & + \end{pmatrix}. \tag{22}$$

On the other hand, the $4 \times 4$ Hadamard matrix

$$W' = \begin{pmatrix} - & + & + & + \\ + & - & + & + \\ + & + & - & + \\ + & + & + & - \end{pmatrix}, \tag{23}$$

with the same initial row was used to construct a $\mathbb{Z}_2$ gauge theory [22]. Both matrices possess combinatorial gauge symmetries corresponding to a $\mathbb{Z}_2$ gauge theory on a square lattice, but (22) has twice as many rows as (23). We will show that the two constructions are equivalent, and $W^{4,2}$ can be reduced to $W'$ procedurally. (Thus we may refer to the latter as $W^{4,2}_{\text{red}}$.)

That the rows of $W^{4,2}_{\text{red}}$ form an incomplete orbit under gauge transformations might suggest that $W^{4,2}_{\text{red}}$ cannot have a combinatorial gauge symmetry. Indeed, multiplying $W^{4,2}_{\text{red}}$ on the right by $R^{12} = \text{diag}\{(-,-,+,+)\}$ we get

$$W^{4,2}_{\text{red}} R = \begin{pmatrix} + & - & + & + \\ - & + & + & + \\ - & - & - & + \\ - & - & + & - \end{pmatrix}, \tag{24}$$

which not only permutes the rows of $W^{4,2}_{\text{red}}$ but also flips the signs of two of them. However, if we relax the restriction that the $L$ matrices be permutations, but also allow for $-1$ in addition to 1 in its non-zero entries, i.e., by letting $L$ be monomial matrices, we can recover the combinatorial gauge symmetry. Physically, this corresponds to not only permuting, but also flipping the matter spins. This operation commutes with a transverse field, so the $\mathbb{Z}_2$ gauge symmetry is preserved in the full Hamiltonian. What makes this manipulation possible is that while $W^{4,2}_{\text{red}}$ does not include every element of the orbit under the action of the gauge transformations, it does include all such entries up to overall signs, so that the action of an $R$ matrix permutes the rows of $W^{4,2}_{\text{red}}$ up to signs, which can be compensated for by the signs of the entries in the $L$ matrix. In this particular case, the corresponding $L$ matrix is

$$L^{12}_{\text{red}} = \begin{pmatrix} 0 & 1 & 0 & 0 \\ 1 & 0 & 0 & 0 \\ 0 & 0 & 0 & -1 \\ 0 & 0 & -1 & 0 \end{pmatrix}. \tag{25}$$

The full and reduced versions of the coupling matrices, $W^{4,2}$ and $W^{4,2}_{\text{red}}$, are mapped to each other as follows. To reconstruct $W^{4,2}$, append to each row $r_a \in W^{4,2}_{\text{red}}$ a row $-r_a$. Conversely, observe that $W^{4,2}$ consists of four pairs of rows that differ by an overall sign, so by choosing one representative from each pair, we form $W^{4,2}_{\text{red}}$. This procedure also produces a correspondence between the full and reduced versions of $L$ matrices. A reduced $L$ matrix is augmented into the full version by replacing each entry by a $2 \times 2$ block. Zeros and ones become zero and identity matrices, while $-1$ becomes a $2 \times 2$ permutation matrix that swaps the two rows that differ by an overall sign. For example, from $L^{12}_{\text{red}}$ in (25), we get

$$L^{12} = \left( \begin{array}{c|cc|c|c} 0 & \begin{matrix} 1 & 0 \\ 0 & 1 \end{matrix} & & 0 & 0 \\ \hline \begin{matrix} 1 & 0 \\ 0 & 1 \end{matrix} & 0 & & 0 & 0 \\ \hline 0 & 0 & & 0 & \begin{matrix} 0 & 1 \\ 1 & 0 \end{matrix} \\ \hline 0 & 0 & & \begin{matrix} 0 & 1 \\ 1 & 0 \end{matrix} & 0 \end{array} \right). \tag{26}$$

Compare the transformation of the full $W$ matrix under the action of $R^{12}$,

$$
WR^{12} = \begin{pmatrix}
+ & - & + & + \\
- & + & - & - \\
- & + & + & + \\
+ & - & - & - \\
- & - & - & + \\
+ & + & + & - \\
- & - & + & - \\
+ & + & - & +
\end{pmatrix}.
\tag{27}
$$

This is a permutation of the rows 1 and 3, 2 and 4, 5 and 8, and 6 and 7, which precisely represented by $L^{12}$. To reduce from a full $L$ matrix, we first sort the rows of the coupling matrix so that rows related by a uniform gauge transformation are adjacent to each other, so that the full $L$ matrix will take a block form. Using a permutation representation of $\mathbb{Z}_2$, each block in the full $L$ matrix is mapped to $+1$ or $-1$.

## 5.2 Tuning the spectrum of a CGS interaction

Next we shall analyze the case that has been studied in [23], namely a $\mathbb{Z}_3$ gauge theory on a triangular lattice (see Appendix A for the change in underlying lattice) in the context of the general method of building a CGS model as well as techniques for reducing the number of matter degrees of freedom and adjusting the spectrum.

A plaquette in a triangular lattice has coordination number 3 and we are working with the gauge group $\mathbb{Z}_3$, so starting from the initial row $(1, 1, 1)$, we construct a $W$ matrix

$$
W^{3,3}[(1,1,1)] = \begin{pmatrix}
1 & 1 & 1 \\
\omega & \omega & \omega \\
\overline{\omega} & \overline{\omega} & \overline{\omega} \\
1 & \omega & \overline{\omega} \\
\omega & \overline{\omega} & 1 \\
\overline{\omega} & 1 & \omega \\
1 & \overline{\omega} & \omega \\
\omega & 1 & \overline{\omega} \\
\overline{\omega} & \omega & 1
\end{pmatrix},
\tag{28}
$$

where $\omega = e^{2\pi i/3}$. Note that the rows can be partitioned into three sets, and within each set the rows differ by an overall phase factor. Following the reduction procedure illustrated by the previous example, we pick one representative from each set to construct a reduced coupling matrix,

$$
W^{3,3}_{\text{red}}[(1,1,1)] = \begin{pmatrix}
1 & 1 & 1 \\
1 & \omega & \overline{\omega} \\
1 & \overline{\omega} & \omega
\end{pmatrix},
\tag{29}
$$

and take the matter degrees of freedom to be $\mathbb{Z}_3$ clock variables.

However the ground state of this CGS Hamiltonian does not lie in the $\mathbb{Z}_3$ flux 0 sector, instead it is in the flux 1 and flux 2 sector, which can be verified from equation (20). To fix this issue, we can adjust the initial row of the $W$ matrix, for example to $(-1, 1, 1)$. An alternative solution is presented in the appendix of [23], that is, a $W$ matrix that combines two orbits

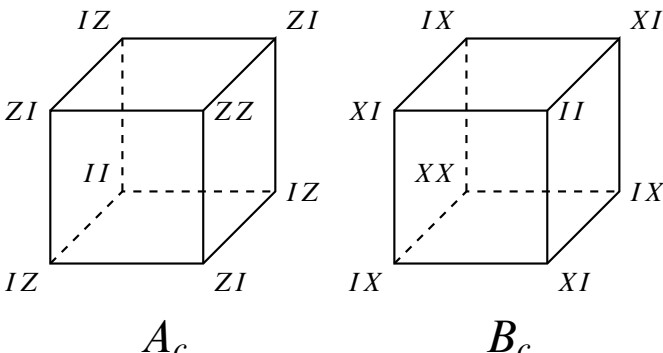

Figure 3: Illustration of the two types of operators in the Hamiltonian of the Haah's code.

under the gauge transformations,

$$W^{3,3}_{\mathrm{red}}[(1,1,\omega),(1,1,\overline{\omega})] = \begin{pmatrix} 1 & 1 & \omega \\ 1 & \omega & 1 \\ 1 & \overline{\omega} & \overline{\omega} \\ 1 & 1 & \overline{\omega} \\ 1 & \overline{\omega} & \omega \\ 1 & \omega & \omega \end{pmatrix} . \tag{30}$$

In Sec. 4 we noted that as long as the rows of $W$ forms an invariant set under the action of the gauge transformations, it possesses a combinatorial gauge symmetry. The simplest invariant sets are orbits of a single row, but they can also combine multiple distinct orbits. This is not obviously advantageous, because the size of such a $W$ matrix is not minimal. However, the above $W$ for the $\mathbb{Z}_3$ CGS turns out to have the correct flux state as the ground state, while the simple $W$ matrix formed out of the orbit of $(1,1,1)$ does not. Thus we have another method of adjusting the spectrum of CGS Hamiltonians.

### 5.3 Gauge theories with more complex structure

To illustrate the power of the CGS construction, we now construct a model with the symmetry of Haah's code. Recall that Haah's code is defined on three-dimensional cubic lattice, where there are two spins on each site. The Hamiltonian is the sum of two types of generators of a stabilizer code, each supported on each cubic cell, $c$, of the lattice,

$$H_{\mathrm{HC}} = -\sum_c A_c - \sum_c B_c .$$

Each operator $A_c$ and $B_c$ is a product of $\sigma^z$'s and $\sigma^x$'s, respectively, acting on a subset of the eight spins on the cubic cell, as illustrated in Fig. 3. Reformulated as a gauge theory, we may treat the $A$'s as charge operators and the $B$'s a generators of the gauge symmetry. Thus to construct a CGS model with the same type of symmetry, we need to compute the group of local gauge transformations on an $A$-type operator generated by conjugating it by $B$-type operators. This is computed in Appendix D. Despite the complex arrangement of sites, the local group of gauge symmetries is very simple, $\mathbb{Z}_2^7$. This means that we could emulate an $A$-type operator by a CGS model of the $(2,8)$-type.

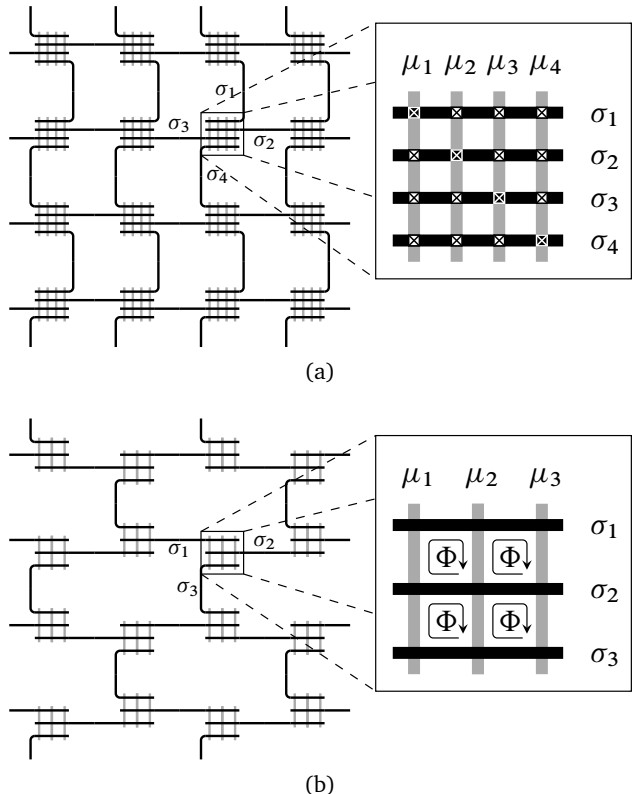

(a)

(b)

Figure 4: Superconducting wire arrays that implement combinatorial gauge symmetries for (a) a $\mathbb{Z}_2$ gauge theory on a square lattice [25] and (b) a $\mathbb{Z}_3$ gauge theory on a honeycomb lattice [23]. (a) The array is a grid of "waffles" in which horizontal wires extending across neighboring waffles (in black) represent the gauge degrees of freedom, and the vertical wires (in gray) are the matter degrees of freedom. Within each waffle, the wires are coupled by Josephson junctions with no phase shift that corresponding to +1 in the coupling matrix (black cross on white square) and junctions with a $\pi$ phase shift that correspond to a $-1$ in the coupling matrix (white cross on black square). (b) In the $\mathbb{Z}_3$ case, the grid of waffles is constructed by the same principle, but within each waffle, the wires are coupled via Josephson junctions, and the phase shift required by the CGS Hamiltonian is realized by a uniform magnetic field that generates a flux of $\Phi = (n + \frac{1}{3})\Phi_0$ within each elementary plaquette of the waffle, where $\Phi_0 = h/2e$ is the flux quantum.

# 6 Implementations

Proposals have been made for realizing a $\mathbb{Z}_2$ CGS model on a square lattice with a superconducting circuit [25], and a classical $\mathbb{Z}_2$ spin liquid with the same combinatorial gauge symmetry was built and observed [26]. These implementations suggest generalizations. Following the approach in [25], superconducting wires can be arranged into an array of "waffles", or grids of intersecting wires, as illustrated in Fig. 4(a). Generally, such wire arrays are described by the Hamiltonian (see also [23])

$$H = H_J + H_C \,,$$

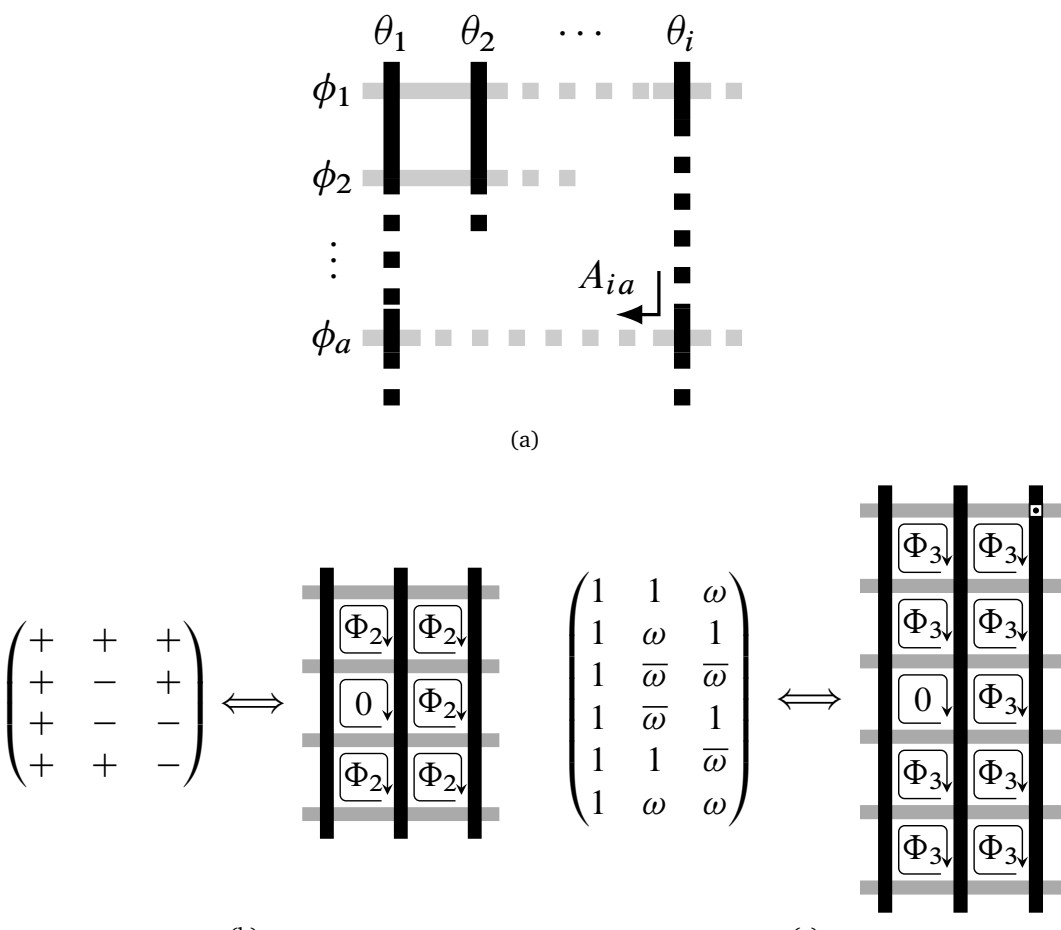

Figure 5: Flux-based superconducting wire array realizations. The general configuration of gauge and matter wires in a waffle and the phase shift through a Josephson junction induced by a magnetic field is depicted in (a). Correspondences between CGS coupling matrices and flux configurations are shown in (b) for the $\mathbb{Z}_2$ CGS model on a triangular lattice and a waffle with a uniform flux $\Phi_2 = \frac{1}{2}\Phi_0$ except for one plaquette, and (c) for the $\mathbb{Z}_3$ CGS theory with the coupling matrix $W_{\mathrm{red}}^{3,3}[(1,1,\omega),(1,1,\overline{\omega})]$ and a waffle with a uniform flux $\Phi_3 = \frac{1}{3}\Phi_0$ except for one plaquette and one junction (represented by a dotted box) that has a $\frac{2\pi}{3}$ phase shift.

with the Josephson coupling

$$H_J = -E_J \sum_p \left[ \sum_{a,i\in p} W_{ai} e^{i(\theta_i - \phi_a)} + \mathrm{h.c.} \right],$$

and a quadratic capacitance term $H_C$ that depends on the conjugate charges to the phase variables $\theta_i, \phi_a$. When the coupling coefficients $W_{ai}$ are those of a CGS model, the Josephson energy possesses a combinatorial gauge symmetry, where the action of gauge groups corresponds to shifts in the location of the minima of Josephson energies. (One can ensure that $H_C$ does not break the symmetry by requiring permutation invariance of the capacitance matrix.) In the regime where $H_J$ is dominant, the minima of the Josephson energy correspond to the minima of the CGS model.

To get the phase factors $W_{ai}$ in the Josephson Hamiltonian, two strategies are available. The junctions can be built as Josephson junctions or $\pi$-Josephson junctions [27–30] when the

$W$ matrix consists only of $+1$ and $-1$ entries, or by a more general approach given in [23], shown in Fig. 4(b), where the $W$ matrix can be realized by threading a uniform flux of $\frac{1}{3}$ flux quantum in every elementary plaquette of the waffle. The latter strategy can in fact be applied to other CGS Hamiltonians.

Generically, the Hamiltonian of a waffle of superconducting wires is the sum of the Josephson energies at the intersections of the wires, whose minima will acquire phase shifts in the presence of a magnetic field. Let $\theta_i$ be the superconducting phase of the wire representing the $i$-th gauge variable, and let $\phi_a$ be the phase of the $a$-th matter wire, both measured at the edges of the waffle (the left edge for $\phi_a$ and the top edge for $\theta_i$). The Josephson junction energy at the intersection of the $\theta_i$ and $\phi_a$ wires is equal to

$$H_{ai} = -E_J \cos(\theta_i - \phi_a + A_{ai}) \,, \tag{31}$$

where

$$A_{ai} = \frac{2\pi}{\Phi_0} \left( \int_{(x_i,y_1)}^{(x_i,y_a)} \mathbf{A} \cdot d\mathbf{l} - \int_{(x_1,y_a)}^{(x_i,y_a)} \mathbf{A} \cdot d\mathbf{l} \right), \tag{32}$$

is the phase accumulated by integrating the vector potential $\mathbf{A}$, which originates from magnetic fluxes, along the path from the top of gauge wire $i$ to the left of matter wire $a$ (See Fig. 5(a)), and $\Phi_0$ is the flux quantum.

A gauge transformation $\mathbf{A} \mapsto \mathbf{A} + \boldsymbol{\nabla}\varphi$ shifts the value of $A_{ai}$, up to a factor of $2\pi/\Phi_0$, by

$$\varphi(x_i,y_a) - \varphi(x_i,y_1) - \varphi(x_i,y_a) + \varphi(x_1,y_a) = -\varphi(x_i,y_1) + \varphi(x_1,y_a) \,. \tag{33}$$

Since these two terms on the right-hand side of (33) only depend on the position where $\theta_i$ and $\phi_a$ are measured, they can be absorbed into $\theta_i$ and $\phi_a$, respectively. Thus fixing a gauge amounts to choosing the reference point for the phases in the matter and gauge wires. We choose the gauge such that

$$\int_{(x_1,y_1)}^{(x_1,y_a)} \mathbf{A} \cdot d\mathbf{l} = \int_{(x_1,y_1)}^{(x_i,y_1)} \mathbf{A} \cdot d\mathbf{l} = 0 \,, \tag{34}$$

in which case $A_{ai}$ is tied to the total magnetic flux $\Phi_{ai}$ within the area enclosed by the left and top edges of the waffle and gauge wire $i$ and matter wire $a$:

$$A_{ai} = 2\pi \frac{\Phi_{ai}}{\Phi_0} \,. \tag{35}$$

These shifts in the Josephson energy minima can be used to implement the coupling matrix $W$ of a CGS model. To obtain the most general $W$ matrix, all junctions are regular Josephson junctions with the exception of those involving the top wire of the waffle. These latter junctions need to be tunable, so that the Josephson phase is that of $W_{1i}$. This can be achieved using the asymmetric DC SQUID design in [25]. Then an appropriate magnetic field should be applied, so that at the junction between the $\theta_i$ and $\phi_a$ wires a phase shift $\Phi_{ai}$ is induced, satisfying

$$W_{ai} = e^{i2\pi \frac{\Phi_{ai}}{\Phi_0}} W_{1i} \,. \tag{36}$$

This relation gives us a one-to-one correspondence between $W$ matrices and flux configurations in waffles, because excluding the first column and first row, an $n \times m$ coupling matrix contains $(n-1)\times(m-1)$ entries, and there are precisely $(n-1)\times(m-1)$ elementary plaquettes in a waffle with $m$ gauge wires and $n$ matter wires.

For example, the $\mathbb{Z}_2$ triangular CGS coupling matrix (4) corresponds to the following flux configuration.

$$\Phi^{2,3} = \frac{1}{2}\Phi_0 \begin{array}{|c|c|} \hline 1 & 0 \\ \hline 0 & 1 \\ \hline 1 & 0 \\ \hline \end{array} \ ,$$

where $\Phi_0 = h/2e$ is the flux quantum. Note that a coupling matrix with its rows reordered or its rows transformed according to an internal symmetry of the matter degrees of freedom will have the same combinatorial gauge symmetry and spectrum, but in general it will correspond to a different flux configuration in the waffle realization. For example, if we swap the second and third rows of (4) and flip the sign of the second and last row of the resulting matrix, the coupling matrix will correspond to the following flux arrangement (as illustrated in 5(b)),

$$\Phi^{2,3'} = \frac{1}{2}\Phi_0 \begin{array}{|c|c|} \hline 1 & 1 \\ \hline 0 & 1 \\ \hline 1 & 1 \\ \hline \end{array} \ .$$

Similarly, while the waffle corresponding to (30) is threaded by $\frac{1}{3}\Phi_0$ and $\frac{2}{3}\Phi_0$ fluxes, switching the fourth and fifth rows allows us to implement it as (see also Fig 5(c)).

$$\Phi^{3,3} = \frac{1}{3}\Phi_0 \begin{array}{|c|c|} \hline 1 & 1 \\ \hline 1 & 1 \\ \hline 0 & 1 \\ \hline 1 & 1 \\ \hline 1 & 1 \\ \hline \end{array} \ .$$

Given the total Josephson energy of a waffle

$$H^{\text{waf}} = -E_J \sum_p \left[ \sum_{i,a \in p} W_{ai}\, e^{i(\theta_i - \phi_a)} + \text{h.c.} \right] , \tag{37}$$

that corresponds to a CGS model with $W$ as its coupling matrix, the minimum is achieved at

$$E_0^{\text{waf}} = -2E_J \sum_a \left| \sum_i W_{ai}\, e^{i\theta_i} \right| , \tag{38}$$

by a phase configuration $(\theta_1, \ldots, \theta_q)$ on the gauge wires that corresponds to the ground state of the CGS Hamiltonian. (Compare to (20).) And as argued in [23], the phases $(\phi_1, \ldots, \phi_N)$ on the matter wires are tethered to the $\theta_i$'s via

$$e^{i\phi_a} = \frac{\sum_i W_{ai}\, e^{i\theta_i}}{\left| \sum_i W_{ai}\, e^{i\theta_i} \right|} , \tag{39}$$

in the same way that the signs of the matter spins in the ground state are fixed by the configuration of the gauge spins. This shows that the ground state degeneracy of the waffle system corresponds exactly to that of the CGS Hamiltonian, so when the Josephson energy is dominant, the superconducting wire array will exhibit a topological phase described by the CGS Hamiltonian.

# 7 Summary and outlook

We have presented a general method for constructing a class of one-and two-body-interaction Hamiltonians with an *exact* gauge symmetry for *any* given Abelian group. The work systematically encompasses earlier sporadic examples of combinatorial gauge symmetry. We illustrated the principles and the construction for a variety of groups and lattice geometries.

The core physical concept was to build a lattice with matter and gauge degrees of freedom, which can be spins or superconducting wires. The core mathematical concept was to classify the allowed interactions between the matter and gauge degrees of freedom. This problem reduced to ensuring consistency between the interaction matrix and action of the gauge group on the underlying quantum variables.

The Hamiltonians with the exact gauge symmetry should contain gapped topological phases for a wide range of parameters. We are hopeful that the construction here is sufficiently general that one or more of these examples can lead to realizations of topologically ordered quantum spin liquids in engineered or programmable quantum devices. At the very least, the mathematical construction here may offer a direction for further theoretical explorations.

## Acknowledgments

We thank Garry Goldstein for insightful discussions as well as for valuable input.

**Funding information** The work of H.Y. and C.C. is supported by the DOE Grant DE-FG02-06ER46316 (work on designing topological phases of matter and foundations of combinatorial gauge theory) and by the NSF Grant DMR-1906325 (work on interfaced topological states in superconducting wires).

## A Lattice gauge theory conventions

In this section we make a comment on the lattice gauge theory convention adopted in this paper and contrast it with the conventions in previous work on combinatorial gauge theory [22,23]. Briefly, in previous examples of $\mathbb{Z}_2$ and $\mathbb{Z}_3$ combinatorial gauge theories, the flux variables of the gauge theory are located on vertices, so the flux operators are supported on stars, of a square lattice and a honeycomb lattice, respectively. Under the conventions of this paper, these constructions would be for a square plaquette-based $\mathbb{Z}_2$ flux operator, and a triangular plaquette-based $\mathbb{Z}_3$ flux operator.

In 2D Abelian lattice gauge theories, the flux and charge variables are dual to each other. Under a transformation that maps each face of graph to a vertex of its dual graph and vice versa, the flux variables can be interpreted as the charge variable of a theory defined on the dual graph. This means that, in terms of physical properties, it makes no difference if the flux variables are supported on the plaquettes of a graph or on the vertices. The more popular convention is to put flux variables on plaquettes, which is also the convention adopted in this paper. See Fig. 6 for an illustration of the $\mathbb{Z}_2$ gauge theory in Section 2. Note that the position of the degrees of freedom and the Hamiltonian do not depend on the underlying lattice chosen.

We also note that the convention of situating flux variables on vertices of the lattice is the source of the terminology "matter" degrees of freedom. These matter variables are directly coupled to the gauge degrees of freedom, and are analogously supported on the vertices of the lattice. While strictly speaking individual "matter" variables don't carry representations of the gauge group, their Hilbert spaces may be rearranged into a direct sum of subspaces that

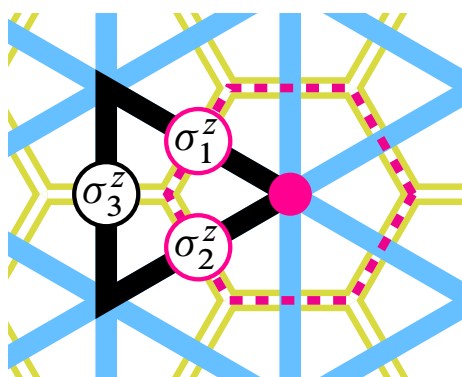

Figure 6: A $\mathbb{Z}_2$ gauge theory on a triangular lattice where the flux operators are supported on plaquettes (blue lattice) can be reinterpreted as a theory on the honeycomb lattice with flux operators supported on the vertices (yellow lattice).

do. Regardless of the true nature of these matter variables, in this paper we are interested in the confined phase of the gauge theory, which is most straightforwardly achieved when the matter variables are integrated out, so we need not dwell on their role in the lattice gauge theory.

## B Further examples of CGS coupling matrices

Here we show two further examples of CGS coupling matrices.

For a $\mathbb{Z}_5$ gauge theory on a triangular lattice, the following coupling matrix is constructed from the initial row $(1, 1, 1)$,

$$
W^{5,3}[(1,1,1)] = \begin{pmatrix}
1 & 1 & 1 \\
1 & \zeta & \zeta^4 \\
1 & \zeta^2 & \zeta^3 \\
1 & \zeta^3 & \zeta^2 \\
1 & \zeta^4 & \zeta \\
\zeta & \zeta^4 & 1 \\
\zeta & 1 & \zeta^4 \\
\zeta & \zeta & \zeta^3 \\
\zeta & \zeta^2 & \zeta^2 \\
\zeta & \zeta^3 & \zeta \\
\zeta^2 & \zeta^3 & 1 \\
\zeta^2 & \zeta^4 & \zeta^4 \\
\zeta^2 & 1 & \zeta^3 \\
\zeta^2 & \zeta & \zeta^2 \\
\zeta^2 & \zeta^2 & \zeta \\
\zeta^3 & \zeta^2 & 1 \\
\zeta^3 & \zeta^3 & \zeta^4 \\
\zeta^3 & \zeta^4 & \zeta^3 \\
\zeta^3 & 1 & \zeta^2 \\
\zeta^3 & \zeta & \zeta \\
\zeta^4 & \zeta & 1 \\
\zeta^4 & \zeta^2 & \zeta^4 \\
\zeta^4 & \zeta^3 & \zeta^3 \\
\zeta^4 & \zeta^4 & \zeta^2 \\
\zeta^4 & 1 & \zeta
\end{pmatrix} .
\tag{40}
$$

Here $\zeta = e^{2\pi i/5}$ is one of the fifth roots of unity. The group of gauge transformations are generated by

$$R^{12} = \text{diag}\{\zeta, \zeta^4, 1\} \,,$$
$$R^{23} = \text{diag}\{1, \zeta, \zeta^4\} \,,$$

and their corresponding $L$ matrices,

$$L^{12} = \text{perm}\{(1,6,11,16,21)(2,7,12,17,22)$$
$$(3,8,13,18,23)(4,9,14,19,24),(5,10,15,20,25)\} \,,$$

and

$$L^{12} = \text{perm}\{(1,2,3,4,5)(6,7,8,9,10)$$
$$(11,12,13,14,15)(16,17,18,19,20)(21,22,23,24,25)\} \,,$$

where $\text{perm}\{\sigma\}$ denotes the permutation matrix corresponding to the permutation $\sigma$.

For a $\mathbb{Z}_2$ gauge theory on a triangular lattice, using the initial row $(1,1,1,1,1,1)$ we can construct a $2^{6-1} \times 6 = 32 \times 6$ CGS coupling. However, since $\gcd(2,6) = 2$, we can reduce the number of rows of this matrix by a factor of 2, to the following $16 \times 6$ matrix,

$$W^{2,6}_{\text{red}}[(1,1,1,1,1,1)] = \begin{pmatrix} + & + & + & + & + & + \\ + & + & + & + & - & - \\ + & + & + & - & - & + \\ + & + & + & - & + & - \\ + & + & - & - & + & + \\ + & + & - & - & - & - \\ + & + & - & + & - & + \\ + & + & - & + & + & - \\ + & - & - & + & + & + \\ + & - & - & + & - & - \\ + & - & - & - & - & + \\ + & - & - & - & + & - \\ + & - & + & - & + & + \\ + & - & + & - & - & - \\ + & - & + & + & - & + \\ + & - & + & + & + & - \end{pmatrix} \,. \tag{41}$$

The generators of the gauge transformation can be taken to be

$$R^{12} = \text{diag}\{-,-,+,+,+,+\} \,,$$
$$R^{23} = \text{diag}\{+,-,-,+,+,+\} \,,$$
$$R^{34} = \text{diag}\{+,+,-,-,+,+\} \,,$$
$$R^{45} = \text{diag}\{+,+,+,-,-,+\} \,,$$
$$R^{56} = \text{diag}\{+,+,+,+,-,-\} \,,$$

and

$$L^{12} = -\text{perm}\{(1,6)(2,5)(3,8)(4,7)(9,14)(10,13)(11,16)(12,15)\} \,,$$
$$L^{23} = \text{perm}\{(1,9)(2,10)(3,11)(4,12)(5,13)(6,14)(7,15)(8,16)\} \,,$$
$$L^{34} = \text{perm}\{(1,5)(2,6)(3,7)(4,8)(9,13)(10,14)(11,15)(12,16)\} \,,$$
$$L^{45} = \text{perm}\{(1,3)(2,4)(5,7)(6,8)(9,11)(10,12)(13,15)(14,16)\} \,,$$
$$L^{56} = \text{perm}\{(1,2)(3,4)(5,6)(7,8)(9,10)(11,12)(13,14)(15,16)\} \,.$$

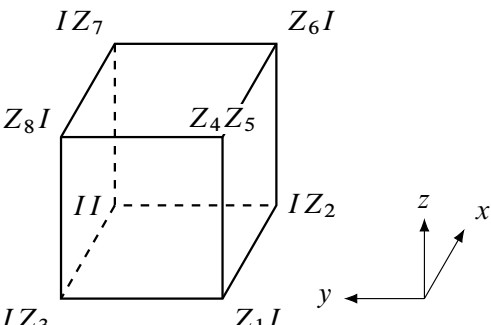

Figure 7: Notations used for writing down the generators of the local group of gauge symmetries. The sites that are acted on non-trivially by type-*A* operators are numbered from 1 through 8 as in the diagram. The type-*B* operators that generate non-trivial transformations of one particular *A*-operator are labeled by their coordinates relative to it, in the coordinate system illustrated on the right.

This example corresponds to the color code [31] on a honeycomb lattice.

## C Breaking spurious degeneracies for $\mathbb{Z}_2$ CGS theories

For a $\mathbb{Z}_2$ CGS theory on a lattice with an odd coordination number $q$, the basic construction outlined in Sec. 4 gives us a $2^{q-1} \times q$ $W$ matrix. The rows of $W$ consist of $q$-element vectors of $\pm 1$ that contain an even number of $-1$'s. To find the ground state of this CGS Hamiltonian, we need only minimize the energy under the constraint $\sigma = 1$, since all other flux 0 states are gauge symmetry partners of this state, and the flux 1 states can be obtained by a global spin flip. In this case, the Hamiltonian reduces to $-\sum_a \mu_a \sum_i W_{ai}$. This is minimized when each $\mu_a$ takes the sign that makes $\mu_a \sum_i W_{ai}$ negative, which is in turn determined by the sign of $\sum_i W_{ai}$. Among these, there are $\binom{q}{2j}$ rows with $2j$ negative signs. When $2j < q/2$, the sum of all entries in the row is positive, otherwise it is negative. Thus in the ground state, the total $\mu$-magnetization is

$$\sum_{j=0}^{\lfloor (q-1)/4 \rfloor} \binom{q}{2j} - \sum_{j=\lceil (q-1)/4 \rceil}^{(q-1)/2} \binom{q}{2j}.$$

Note that the first sum is over the even binomial coefficients up to $(q-1)/2$, and the latter sum is over the remaining even binomial coefficients. Since $\binom{q}{2j} = \binom{q}{q-2j}$ and $q$ is odd, the latter sum is equivalent to a sum over the odd binomial coefficients up to but not including the $(q-1)/2$-th one. This means that we can rewrite the total $\mu$-magnetization as

$$\sum_{j=0}^{(q-1)/2} (-1)^j \binom{q}{j}.$$

A simple inductive argument shows that this is equal to

$$(-1)^{(q-1)/2} \binom{q-1}{(q-1)/2}.$$

This shows that the $\mu$-magnetization in the CGS Hamiltonian for $k = 2$ and $q$ odd is always non-zero, thus guaranteeing that a uniform field on the matter spins will serve to split the ground state degeneracy between flux 0 and flux 1 states.

Table 1: Generators of the group of local gauge symmetries.

| Generator | Decomposition | |
|---|---|---|
| $P_{12}$ | $B_{(0,-1,-1)}$ | |
| $P_{23}$ | $B_{(0,-1,-1)} \cdot B_{(-1,0,-1)}$ | |
| $P_{34}$ | $B_{(-1,0,0)}$ | |
| $P_{45}$ | $B_{(-1,-1,1)}$ | |
| $P_{56}$ | $B_{(0,-1,1)}$ | |
| $P_{67}$ | $B_{(1,0,1)}$ | |
| $P_{78}$ | $B_{(0,1,1)}$ | |

# D  Group of local gauge symmetries of the *A*-type operators in Haah's code

To describe the local group of gauge symmetries $\mathcal{G}_A$ of a type-*A* operator in the Haah's code Hamiltonian, it is convenient to label sites that it acts on as in Fig. 7. The gauge symmetry

generated by a type-*B* operator can then be written as the a multiplication by a matrix *R*,

$$\begin{pmatrix} Z_1 \\ \vdots \\ Z_8 \end{pmatrix} \xleftarrow{B[-]B^{-1}} R \begin{pmatrix} Z_1 \\ \vdots \\ Z_8 \end{pmatrix} .$$

Since *B* is a product of Pauli *x*-operators, the matrices *R* are diagonal matrices with ±1 along the diagonal. Thus for *A* to be invariant under this transformation, there must be an even number of −1 along the diagonal. In other words, the local group of gauge symmetries must be a subgroup of the group consisting of operations flipping an even number of the spins in $(Z_1, \ldots, Z_8)$, which we shall denote by $\tilde{\mathcal{G}}$. A generating set of $\tilde{\mathcal{G}}$ can be taken to be $\{P_{i,i+1}\}_{i=1,\ldots,7}$, where $P_{i,i+1}$ can be represented by the diagonal matrix

$$R_{i,i+1} = \text{diag}\{+1, \ldots, +1, \overset{i\text{-th}}{-1}, \overset{(i+1)\text{-th}}{-1}, +1, \ldots, +1\} .$$

This also shows that $\tilde{\mathcal{G}} \cong \mathbb{Z}_2^7$. Now by showing that every element in this generating set can be expressed in terms of symmetry operations generated by type-*B* operators, we can prove that the local group of symmetries $\mathcal{G}_A$ is exactly equal to $\tilde{\mathcal{G}}$. The generators of $\tilde{\mathcal{G}}$ and their decompositions in terms of *B*-operators are tabulated in Table 1. Note that there are 14 type-*B* operators that generate non-trivial symmetries of *A*, so this computation also shows that they are not all independent of each other.

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
