# Peer review of "Abelian combinatorial gauge symmetry"

_SciPost Physics, doi:SciPost Phys. Core 7, 014 (2024)_

## Round 1 · Referee Report · Anonymous (Referee 1) · 2023-3-19

Strengths

1- The paper provides a good review of the concept of combinatorial gauge symmetry (CGS). 2- The authors extend the construction of models with CGS to arbitrary Abelian groups.

Report

CGS is an intriguing concept that has been introduced recently in Ref. 21 with the goal to construct relatively simple spin models which have an exact local symmetry and host topologically ordered phases of matter. In the present manuscript, Yu et al. present a detailed discussion of models with Abelian CGS. The authors review several known examples, generalise theses to arbitrary Abelian groups and classify them. As one application, the Haah's code is interpreted from the point of view of CGS. Finally, possible implementations with superconducting wire arrays are discussed.

The manuscript is well organised and provides a great introduction into CGS. Beyond that the only major new result of this paper is the generalisation to arbitrary Abelian groups, which is rather straightforward. While the manuscript meets all necessary acceptance criteria of SciPost Physics, I think that is does not fulfill one of the expectations:

i) It neither presents a groundbreaking theoretical discovery -- since CGS was already introduced in Ref. 21;

ii) nor does it present a breakthrough on a long-standing problem -- to my knowledge there is no such problem associated with CGS. An obvious question is the possibility to construct simple models with non-abelian gauge groups, but this question is not addressed at all;

iii) nor does it open a new research direction with clear potential for many follow-ups -- again, CGS has already been introduced previously, and the present study rather closes the situation in the context of Abelian groups;

iv) nor does it provide a novel synergetic link among different areas -- the presented re-interpretation of Haahs's code might have the potential to improve our understanding of fractons, but the author's discussion is rather short here.

Nevertheless, I think the manuscript is worth publishing since it advances the general understanding of CGS and provides an excellent introduction to the topic. I therefore recommend publication in SciPost Physics Core when the authors addressed my comments below.

Requested changes

1- I am missing a discussion of the stability of CGS. The authors emphasise that the construction of spin models consisting only of one- and two-body interactions simplifies their implementation. I agree that this is much simpler than directly engineering, e.g., plaquette interactions. However, since any implementation will be faulty, I wonder what happens to the model in Eq. (6) in the presence of errors such a small perturbation to the (fine-tuned) coupling matrix W or disorder in the external fields.

2- Fig. 2 does not seem to be referred to in the main text.

3- A few typographical errors: i) after Eq. (6): "The vectors [..] and so on ARE shorthands .." (verb missing) ii) in the caption of Fig. 4: "Within each waffle, [..] phase shift THAT corresponding to .." and in the last line "walffe".

  • validity: top
  • significance: ok
  • originality: ok
  • clarity: high
  • formatting: excellent
  • grammar: excellent

Author:  Hongji Yu  on 2023-10-03  [id 4022]

(in reply to Report 1 on 2023-03-19)

We thank the referee for their report and valuable suggestions. Below, we address their comments with reference to revisions in our manuscript.

the only major new result of this paper is the generalisation to arbitrary Abelian groups, which is rather straightforward. ... CGS was already introduced in Ref. 21;

The construction of CGS models with arbitrary Abelian groups is not a simple generalization of previously existing work. While the $\mathbb{Z}_2$ and $\mathbb{Z}_3$ CGS models were discovered in previous works, and their properties were understood through the symmetry of the matrix $W$ of coupling constants in the form $L^\dagger W R = W$ dubbed the combinatorial gauge symmetry, there was no systematic method of constructing such matrices. The initial observation in the $\mathbb{Z}_2$ and $\mathbb{Z}_3$ that the coupling matrices are Hadamard matrices turn out to be insufficient to generate other combinatorial gauge theories. Our current work provides the first systematic construction. We use the method of starting with an initial non-gauge symmetric coupling and symmetrize according to the local gauge transformations. This generates a much more general class of Hamiltonians. It is also worth noting that the previously discovered $\mathbb{Z}_2$ and $\mathbb{Z}_3$ models are not direct applications of the general construction either. They require non-obvious choices of initial couplings, as well as reduction, which are discussed.

nor does it present a breakthrough on a long-standing problem -- to my knowledge there is no such problem associated with CGS

It has been a long-standing problem in the study of topological states of matter that while many topological phases have been predicted, their often highly artificial constructions make experimental realizations difficult. Natural materials are unlikely to exhibit these models, and their multi-body interactions are difficult to simulate with existing techniques. We move towards that direction by proposing realistic Hamiltonians with one- and two-body interactions that could possibly be implemented on programmable quantum devices.

nor does it open a new research direction with clear potential for many follow-ups .., the present study rather closes the situation in the context of Abelian groups

As mentioned by the reviewer, an obvious follow-up to Abelian CGS models is the construction of non-Abelian models. This prospect is addressed in the final section of the paper. There are new challenges in this respect, for example how to simulate a non-Abelian gauge degree of freedom under the one- and two-body constraint, as well as formulating a non-Abelian version of the combinatorial gauge symmetry itself. Moreover, this work motivates models that possess CGS but are not stabilizer codes. Other assumptions, such as the finiteness of the gauge group can be altered, which may very well generate new interesting models. More detailed experimental proposals could also follow, beyond the scheme sketched out in our paper. In any case, we conceive of this work not as closing up a project, but rather as laying a foundation, upon which novel and unexpected results can be built.

the presented re-interpretation of Haahs's code might have the potential to improve our understanding of fractons, but the author's discussion is rather short here

We would like to note that the Haah's code example of a CGS model is not a simple rewrite of Haah's Hamiltonian, but a new model that shares the same gauge symmetry. Our model is potentially a two-body fractonic model, which could point to an experimental realization of fractons and provide insight into the class of gauge theories with fractonic excitations that don't allow two-body interactions.

I am missing a discussion of the stability of CGS.

We appreciate the referee's suggestion, and have expanded the relevant sections in the paper to include a discussion of the stability of the topological phase of a CGS model (See [link and cite]). To summarize, the CGS Hamiltonian in Eq (6) is in the topological phase if the couplings obey the energy hierarchy [add condition]. When this is the case, there is a finite gap. As long as a symmetry breaking perturbation is small compared to this gap, in the thermodynamic limit, the splitting between gauge symmetric states is exponentially small in the system size, so the topological phase is preserved.

---

## Round 1 · Referee Report · Anonymous (Referee 2) · 2023-3-20

Strengths

  • This paper addresses the timely question of realising topological phases using sufficiently simple Hamiltonians (at most 2-body interactions)
  • This paper provides a summary of previous work on this topic (Combinatorial gauge symmetry) by several of the same authors.
  • Overall, the paper is well written and sufficiently detailed to follow the arguments and calculations; it furthermore starts with (motivating) example.

Weaknesses

  • I find the presentation/discussion of gauge symmetry (in particular in the motivating example) confusing.
  • The novelty of the current paper is supposed to be in the generalisation to "general abelian groups", but the relevant results are stated rather sloppily (e.g. I believe that the central CGS hamiltonian in Eq 6 is not Hermitian in the way written) and informally.
  • In relation to the previous comment, I think that the paper could be structured much better. General claims are stated in running text in sections treating specific examples, so it is hard to find the main results of this paper.

Report

In this manuscript, the authors present the concept of "combinatorial gauge symmetry", which is a particular way of implementing an exact gauge symmetry using at most 2-body interactions. This approach was developed by some of the authors for the case of a pure $Z_2$ gauge theory in an earlier paper. Furthermore, a different subset of the authors showed how to implement this proposal on a superconducting circuit.

The current paper summarises these results, and extends them to the case of arbitrary abelian groups. Given that any abelian group is a direct product of cyclic groups, there is a short paragraph explaining how to deal with such product groups, and focusses on the cyclic groups for the remainder.

The paper starts with recapitulating the $Z_2$ construction in section II. However, here the paper becomes confusing (at least to me). Having a traditional perspective on gauge theories, the way to interpret the toric code as a gauge theory is to interpret the loop operators as the magnetic energy, whereas the vertex operator implements the Gauss law as an energy penalty (instead of as a hard Hilbert space constraint). While charge and flux are dual variables in the abelian case and can thus be interchanged, I do find it somewhat confusing that this paper takes this dual perspective (only keeping vertex operators, referring to them as magnetic flux, and associating the gauge transform with plaquettes) without mentioning. I find this confusing in particular because for the rest of the paper, gauge transforms are associated with the sites and the new degrees of freedom that are introduced to implement the combinatorial gauge symmetry are also associated to the vertices. Given that these are actually the plaquettes in the way people usually think about gauge theories, I find it somewhat inappropriate to refer to them as matter degrees of freedom. Hence, I believe some discussion about these aspects are in order.

Next, I am confused by how the new theory with the combinatorial gauge symmetry seems to have $(q-1)$ independent symmetry generators associated with every vertex. Does this mean that this construction transforms a pure gauge theory with group $G$ into one with group $G^{(q-1)}$ ? Furthermore, these generators are associated with two neighbouring links along a vertex, which makes a statement such as "The gauge transformations are generated by products of $X = ... $ along a loop." on page 4 very confusing. I don't see what "loop" is being referred to.

Zooming out, I find the structure of the paper quite unfortunate, as it is hard to quickly understand the novel results, as well as to distinguish between general statements and specific examples. For example, the discussion of the k=2, q=4 example in section IV.B contains a more general result stated at the end of this paragraph. However, it is very hard to get a good grasp about which results are general, which are examples (e.g., regarding the strategy for having the correct intended ground state with zero "flux" associated to the "sites"). Beyond the motivating example, I think it would be good to have the main results formulated as clear statements that stand out in the text, maybe even propositions or theorems which are then argued for, and only then illustrated using examples.

Finally, there are some obvious mistakes or issues. The general CGS Hamiltonian in Eq 6 is not Hermitian as states, because the equivalent of $\sigma^x$ in the general case of a cyclic group $Z_k$ is not Hermitian. It is also not trivial wether the first term coupling gauge to matter spins is Hermitian.

Requested changes

1) Add discussion of how to interpret this as a gauge theory, whether the new degrees of freedom truly act as matter, what is the gauge symmetry, etc. 2) Improve the structure of the paper to more clearly present the (novel) results, and make them clear as explicit statements or propositions. 3) Correct obvious mistakes.

  • validity: good
  • significance: good
  • originality: high
  • clarity: good
  • formatting: good
  • grammar: excellent

Author:  Hongji Yu  on 2023-10-03  [id 4023]

(in reply to Report 2 on 2023-03-20)

We thank the referee for their report and valuable suggestions. Below, we address their comments with reference to revisions to our paper.

We have made major revisions to our paper according to the suggestion to highlight the novel results and extract general discussions from examples. The updated draft now contains a new section (Section 3) that summarizes the results. A number of high level discussions about modifications to CGS Hamiltonians in particular cases have been extracted from the specific examples and organized into the main theoretical section (Section 4).

We would also like to address the following points more specifically.

While charge and flux are dual variables in the abelian case and can thus be interchanged, I do find it somewhat confusing that this paper takes this dual perspective

The choice of associating flux variables with vertices rather than plaquettes was a choice inherited from the CGS construction of the $\mathbb{Z}_2$ gauge theory on a square lattice (toric code), where the stars and plaquettes are exactly dual. Indeed, as the referee has noted, in the Abelian case flux and charge can be exchanged, so the choice is mainly a matter of convention.

However, we recognize that this dual view is not the most common convention, and is prone to cause confusion to readers unfamiliar with previous work on CGS models. Therefore, we have opted to switch the role of flux and charge operators back to the more common notation, while noting the break from the existing Z2 and Z3 work in Appendix A, as well as commenting on the term "matter spins".

As a result of this change of convention, we believe we are able to address the following points also,

I find this confusing in particular because for the rest of the paper, gauge transforms are associated with the sites and the new degrees of freedom that are introduced to implement the combinatorial gauge symmetry are also associated to the vertices. ... The gauge transformations are generated by products of X=... along a loop." on page 4 very confusing. I don't see what "loop" is being referred to.

We thank the reviewer for noting the confusion caused by the choice of convention. In the updated draft, we have adopted the standard convention of placing gauge variables (fluxes) on plaquettes, and generators of gauge transformations (charges) on stars.

Originally we referred to gauge transformations supported on closed loops. Now they should be understood as being supported on the domain walls separating the charges that generate them and the rest of the lattice, or "dual loops" in a sense. This is what we referred to by "gauge transformations are generated by products of X=... along a loop" in the first draft.

Next, I am confused by how the new theory with the combinatorial gauge symmetry seems to have $(q−1)$ independent symmetry generators associated with every vertex. ...

To address the question raised here, we have rewritten Section 4 to include a detailed exposition on the various algebraic objects related to the gauge group. Key to the discussion are the gauge group itself, the group of all gauge transformations, and the group of transformations that act on the degrees of freedom in a plaquette operator non-trivially (dubbed "local group of gauge transformations"). The group that has order $|G|^{(q-1)}$ (in some cases) is the last one. Hopefully we have been able to clarify this point in the updated draft.

---

## Round 2 · Author Response

We sincerely thank the referees for their comments and suggestions, and have made substantial revisions to the paper accordingly. We have restructured the paper to highlight the main results, clarified key theoretical arguments, elaborated on experimental realizations, and polished the language and notations. We trust that our work is now ready for publication in SciPost Physics.

Please see below for a list of changes. We have also posted our detailed replies to the referee reports.

---

## Round 2 · List of Changes

The paper has undergone major revisions, including
- A new section summarizing the key results (Section 3),
- Reorganized and expanded main theoretical section, including general results previously derived alongside specific examples (Section 4),
- Simplified and clarified examples (Section 5),
- Elaborations on the superconducting wire array realization (Section 6),
- Adopting the standard lattice gauge theory convention for supports of charge and flux variables (throughout the paper, explained in Appendix A).

---

## Editorial Decision

published